# Structure of the active G$_i$-coupled human lysophosphatidic acid receptor 1 complexed with a potent agonist

Hiroaki Akasaka[1], Tatsuki Tanaka [1], Fumiya K. Sano[1], Yuma Matsuzaki[1], Wataru Shihoya [1]✉ & Osamu Nureki [1]✉

Lysophosphatidic acid receptor 1 (LPA$_1$) is one of the six G protein-coupled receptors activated by the bioactive lipid, lysophosphatidic acid (LPA). LPA$_1$ is a drug target for various diseases, including cancer, inflammation, and neuropathic pain. Notably, LPA$_1$ agonists have potential therapeutic value for obesity and urinary incontinence. Here, we report a cryo-electron microscopy structure of the active human LPA$_1$-G$_i$ complex bound to ONO-0740556, an LPA analog with more potent activity against LPA$_1$. Our structure elucidated the details of the agonist binding mode and receptor activation mechanism mediated by rearrangements of transmembrane segment 7 and the central hydrophobic core. A structural comparison of LPA$_1$ and other phylogenetically-related lipid-sensing GPCRs identified the structural determinants for lipid preference of LPA$_1$. Moreover, we characterized the structural polymorphisms at the receptor-G-protein interface, which potentially reflect the G-protein dissociation process. Our study provides insights into the detailed mechanism of LPA$_1$ binding to agonists and paves the way toward the design of drug-like agonists targeting LPA$_1$.

Lysophospholipids are simple phospholipids that activates GPCRs to evoke signals involved in a broad range of biological processes[1]. They are characterized by a single hydrocarbon chain and a polar head group, which can be divided into two subgroups; molecules containing the glycerol backbone (lysoglycerophospholipids) and the sphingoid base backbone (lysosphingolipids). The representative lysophospholipids in each of the two groups are lysophosphatidic acid (LPA) and sphingosine-1-phosphate (S1P), which activate the LPA receptors (LPA$_{1-6}$) and the S1P receptors (S1P$_{1-5}$), respectively[2]. LPA$_{1-3}$ and S1P$_{1-5}$ belong to the endothelial differentiation gene (EDG) family, based on the amino acid sequence identity, and exhibit conserved structural features in the ligand-binding pocket. By contrast, LPA$_{4-6}$ belong to the non-EDG family, which is more closely related to the purinergic P2Y receptor family[2,3]. Furthermore, phosphate-modified derivatives of LPA exist in vivo to mediate signaling through different GPCRs. For example, lysophosphatidylserine and lysophosphatidylinositol activate the lysophosphatidylserine receptors (LPS$_{1-3}$) and GPR55, respectively[4–6]. A dephosphorylated LPA derivative, 2-arachidonyl glycerol (2-AG), activates cannabinoid receptors (CB$_{1, 2}$), which are most related to the EDG family at the phylogenetic level[7,8]. These diverse lipid-sensing GPCRs precisely discriminate between the chemical structures of lipid ligands[9].

In 1996, LPA$_1$ was the first identified LPA receptor[10], and thus LPA-LPA$_1$ signaling is the best-studied among the LPA receptors[11]. LPA$_1$ couples to the G proteins such as G$\alpha_i$, G$\alpha_q$, and G$\alpha_{12/13}$, and transduces various intracellular signals, e.g., increased Ca$^{2+}$ concentration and actin reorganization by the Rho/ROCK pathway. LPA$_1$ is widely expressed in several organs to control cell proliferation and survival, cell–cell contact, cell migration, and cytoskeletal morphological changes. The essential physiological functions of LPA$_1$ are nervous-system tissue development and chondrocyte differentiation. LPA$_1$ is associated with various diseases such as cancer, inflammation, and

[1]Department of Biological Sciences, Graduate School of Science, The University of Tokyo, Bunkyo, Tokyo 113-0033, Japan. ✉e-mail: wtrshh9@gmail.com; nureki@bs.s.u-tokyo.ac.jp

neuropathic pain, and thus is a pathologically important receptor that is an essential drug target. Because LPA$_1$ signaling promotes cancer progression in many tissues[12], LPA$_1$ antagonists have been well studied as anti-cancer drugs. Moreover, some preclinical studies suggested the potential therapeutic value of selective LPA$_1$ agonists for obesity[13,14] and urinary incontinence[14,15]. However, the metabolic instability of LPA and its resultant short half-life have complicated the functional characterization of supplemented LPA[16]. Nonlipid LPA$_{1–3}$ agonists are poorly reported, and thus identifying new potent and more stable agonists would be useful to explore and consolidate the potential therapeutic benefits of LPA receptors agonistic drugs. To date, the agonist-bound structures of the S1P receptors and CB receptors have been reported, revealing their lipid-ligand recognition mechanisms relevant for LPA$_1$[3,17–24]. While the antagonist-bound LPA$_1$ inactive structure was also reported[24], little is known about how LPA selectively activates the LPA receptors among the lipid-sensing GPCRs, limiting the design of drug-like LPA receptor agonists.

Here we report the 3.5 Å-resolution cryo-electron microscopy (cryo-EM) structure of the human LPA$_1$-G$_i$ signaling complex bound to an LPA analog with more potent activity against LPA$_1$. Close examination of the LPA$_1$ structure reveals the mechanisms of ligand-lipid binding, receptor activation, and G-protein coupling.

## Results

### Overall structure

For the structural study, we developed a chemically stable analog of LPA (Supplementary Method). The glycerol backbone of sn−2 LPA was partially replaced by an amide bond, and the cis-9 double bond in the acyl chain was replaced by an aromatic moiety (Fig. 1a). In a NanoBiT-G-protein dissociation assay[25], the resulting compound

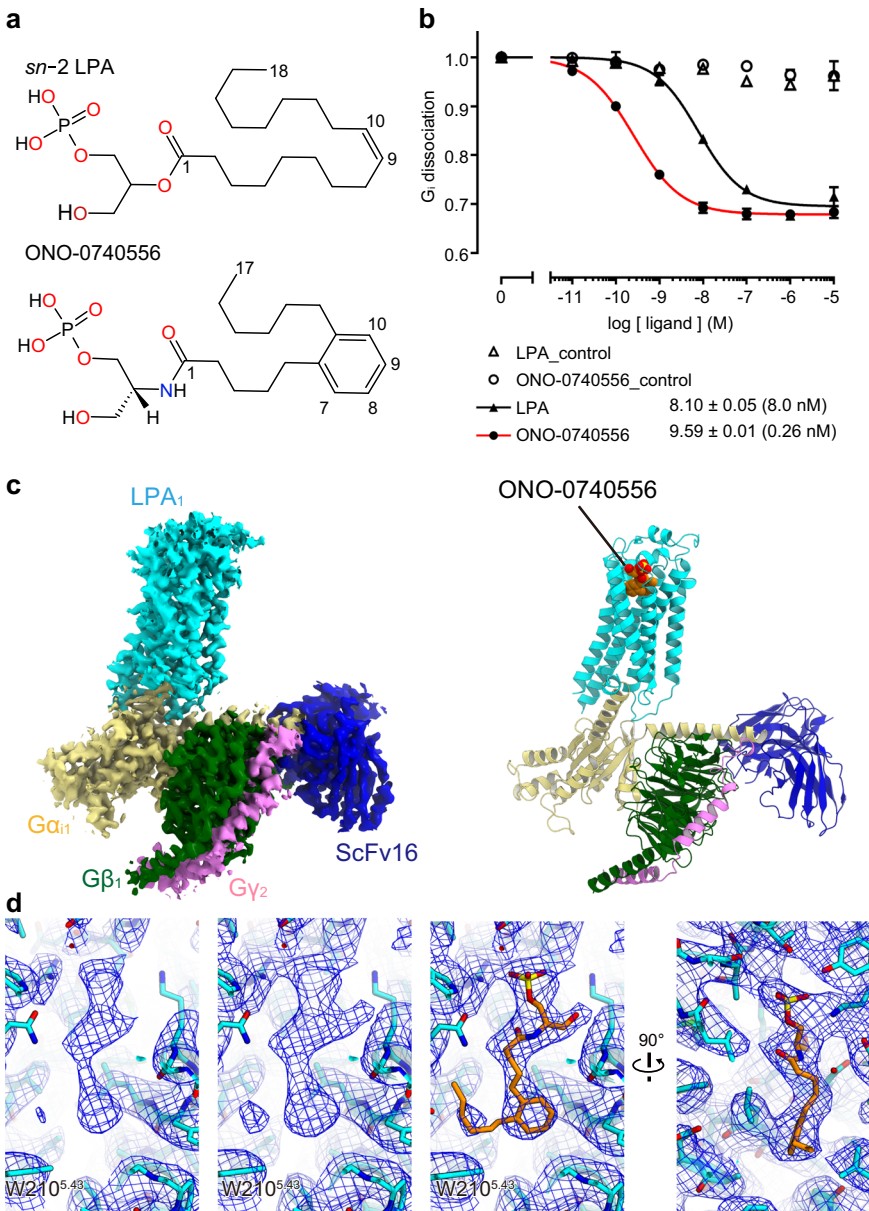

**Fig. 1 | Overall structure of the LPA$_1$ -Gα$_{i1}$β$_1$γ$_2$-ScFv16 complex. a** Chemical structures of LPA and ONO-0740556. **b** Ligand-induced G$_i$ activation by LPA$_1$-G$_i$ activation was measured by the NanoBiT-G-protein dissociation assay. Concentration–response curves are shown as means ± s.e.m. (standard error of the mean) from three independent experiments. Source data are provided as a Source Data file. **c** Sharpened cryo-EM maps and refined structures. **d** Densities around the agonist at different density levels. We observed three strong densities, and assigned the phosphate group, glycerol backbone, and aromatic group of ONO-0740556 to them, given the surrounding environment. Furthermore, we also observed a density above W210$^{5.43}$, so we extended the acyl chain to it.

ONO-0740556 showed agonist activity with an $EC_{50}$ value of 0.26 nM for the human $LPA_1$, which is 30-fold higher than that of LPA (Fig. 1b, Supplementary Fig. 1, Supplementary Table 1, and Supplementary Methods). This result indicates that ONO-0740556 is more suitable for the structural study toward the design of a drug-like LPA receptor agonist.

We independently expressed and purified $LPA_1$, $G_i$ trimer, and scFv16 in insect cells and mixed them, and then purified the complex by anti-Flag affinity chromatography and size exclusion chromatography. The structure of the purified complex was determined by single-particle cryo-EM analysis with an overall resolution of 3.5 Å (PDB 7YU3) (Fig. 1c, Supplementary Fig. 2, Supplementary Table 2, and "Methods"). In this analysis, we subtracted the minimal apparent density for the micelle and the α-helical domain of the $Gα_{i1}$ subunit to consider their flexibilities. The local resolution analysis demonstrated that the interaction site of $Gα_{i1}$, $β_1$, scFv16 and the interface between the $Gα_{i1}$ subunit and the intracellular side of the receptor have higher resolutions. In contrast, the extracellular part of the receptor has a lower resolution (Supplementary Fig. 3). Thus, we performed a refinement with a mask on the receptor, and obtained the receptor structure with a nominal resolution of 3.7 Å (PDB 7YU4) (Supplementary Fig. 3 and Supplementary Table 2). Moreover, in this procedure, the density of ONO-0740556 ligand became more clearly observed within the orthosteric site (Fig. 1d). Based on this structure, we analyzed the modes of agonist binding and receptor activation.

## ONO-0740556 binding site

ONO-0740556 provides an extensive interaction network with N-term, ECL1, 2, and TMs 2, 3, 5, 6, and 7 of the receptor (Fig. 2a–d). The binding site consists of a polar recognition region on the extracellular side and a hydrophobic pocket within the transmembrane region (Fig. 2b–d). This binding manner configuration is also found in S1P receptors[17,21–23]. The head phosphate and glycerol moieties of ONO-0740556 are located in the polar recognition site (Fig. 2a–d). Two oxygen atoms of the head phosphate form salt bridges with $K39^{N-term}$ and $R124^{3.28}$ (superscripts indicate Ballesteros–Weinstein numbers) (Fig. 2a–d). The phosphate group also forms a hydrogen bond with $Y34^{N-term}$, and is tightly recognized by the positively charged resides $K294^{7.36}$. Moreover, the nitrogen atom in the amide bond forms a hydrogen bond with $E293^{7.35}$ (Fig. 2a–d). The agonist binding mode at the polar recognition region is consistent with the previous molecular dynamics simulation[24,26], which revealed that $Y34^{N-term}$ and $K39^{N-term}$ bind the head group, and with the mutational analysis in which the mutations of $Y34^{N-term}$, $K39^{N-term}$ and $R124^{3.28}$ to alanine reduced the responses elicited by ONO-0740556 (Fig. 2e).

By contrast, the long acyl chain fits into the transmembrane pocket in a bent conformation and forms extensive hydrophobic interactions with the receptor (Fig. 2b–d). Notably, the aromatic moiety in the middle of the acyl chain is sandwiched between two leucines, $L278^{6.55}$ and $L297^{7.39}$. Among them, $L297^{7.39}$ forms a CH–π interaction with the moiety. Consistently, both the $L278^{6.55}A$ and $L297^{7.39}A$ mutations reduced the affinity, and $L297^{7.39}A$ had a more pronounced reduction (Fig. 2e). These data indicates that $L297^{7.39}$ plays a critical role in ONO-0740556 binding. Furthermore, in the acyl chain, the C14 carbon forms a CH–π interaction with $W210^{5.43}$ (Fig. 2b–d). The $W210^{5.43}A$ mutant completely lost the response for ONO-0740556, although its expression level was similar to that of wild type, indicating the functional importance of $W210^{5.43}$ for ONO-0740556 binding and receptor activation (Fig. 2e, Supplementary Fig. 4, and Supplementary Table 3). The residues involved in the agonist binding are highly conserved among the EDG family LPA receptors ($LPA_{1–3}$), suggesting a similar mechanism for LPA recognition. However, $K39^{N-term}$ in LPA1 is replaced by $T19^{N-term}$ in $LPA_3$, suggesting that the head phosphate recognition is different between $LPA_1$ and $LPA_3$ (Fig. 2f).

## Structural insight into LPA selectivity

The lysophospholipids LPA and S1P and the dephosphorylated LPA derivative 2-AG can selectively activate the evolutionarily related LPA receptors, the S1P receptors, and CB receptors, respectively[7,8]. To elucidate the mechanism of their lipid preference, we compared the agonist-bound structures of $LPA_1$, $S1P_3$[27], and $CB_1$[28]. Their transmembrane regions superimposed well (Fig. 3a), but the N-terminus of $CB_1$ is different, with only partial structures observed (Fig. 3b). Focusing on the extracellular side of $CB_1$, $F177^{2.64}$ is present at the position occupied by the head phosphates of the agonists in $LPA_1$ and $S1P_3$ (Fig. 3c–e). Thus, lipid ligands lacking phosphate groups selectively activate $CB_1$. A comparison of the phosphate recognition sites in $LPA_1$ and $S1P_3$ revealed that the head phosphate group is in almost identical positions and forms a salt bridge with lysine ($K39^{N-term}$ and $K27^{N-term}$ in $LPA_1$ and $S1P_3$, respectively) and arginine ($R124^{3.28}$ and $R114^{3.28}$ in $LPA_1$ and $S1P_3$, respectively). Moreover, the phosphate group also forms a hydrogen bond with the tyrosine ($Y34^{N-term}$ and $Y22^{N-term}$ in $LPA_1$ and $S1P_3$, respectively) (Fig. 3f and Supplementary Table 4). Thus, $S1P_3$ and $LPA_1$ similarly recognize phosphate groups. Overall, the salt bridges near the lysine and arginine residues enhance the recognition of the head phosphate group in $LPA_1$ and create selectivity for LPA over other lysophospholipid mediators that have modified phosphate groups with weaker negative charges.

We next focused on the hydrophobic pockets accommodating the acyl chain. At the position 5.43, a tryptophan residue creates the bottom of the pocket in $LPA_1$ and $CB_1$ (Fig. 3c, e). The presence of tryptophan in this position only occurs in 1% of all class A GPCRs and is unique to the LPA and CB receptors[24], and it is involved in the agonist binding in both receptors (Fig. 3c, e). The corresponding residue in $S1P_3$ is $C200^{5.43}$, with a smaller side chain (Fig. 3d). This amino-acid difference allows to create a deeper pocket in $S1P_3$ as compared to $LPA_1$ and $CB_1$ (Fig. 3c–e, g). Furthermore, $F119^{3.33}$, $L189^{ECL2}$, $L259^{6.51}$, and $F263^{6.55}$ in $S1P_3$ are replaced by $D129^{3.33}$, $A199^{ECL2}$, $G274^{6.51}$, and $L278^{6.55}$ in $LPA_1$, respectively. As a result, they create a bulge of the hydrophobic pocket toward TM 5–7 in $LPA_1$ (Fig. 3g). These amino-acid replacements allow the hydrophobic pockets of $LPA_1$ and $CB_1$ to be spherical (Fig. 3c–e, g) and thus they can accommodate long and bent unsaturated acyl chains (Fig. 3c), accounting for the fact that $LPA_1$ prefers unsaturated LPA species with a cis-9 double bond in bent shapes (oleic (18:1), linoleic (18:2), and linolenic (18:3))[29]. By contrast, the S1P in the human body has only 18:1, with the trans-4 double bond in a linear configuration. Thus, linear S1P can activate S1P receptors with a deep, linear pocket, in contrast to $LPA_1$ with a shallow, wide pocket. Together, the polar recognition site, which strongly recognizes phosphate groups, and the hydrophobic pocket region, which recognizes an unsaturated acyl chain, contribute to the LPA selectivity by $LPA_1$.

## Receptor activation

To examine the activation mechanism of $LPA_1$, we compared the $LPA_1$ structures in the present agonist-bound active state and the previously-reported antagonist-bound inactive states[24]. On the intracellular side, TM6 is displaced outward by about 8.2 Å, and TM7 is shifted inward by about 4.1 Å. Such structural changes are typical of class A GPCRs and allow G-protein coupling and activation[30] (Fig. 4a, b). At the ligand-binding site, the positively charged residues $K39^{N-term}$ and $R124^{3.28}$ similarly recognize the negative charges in both antagonists and agonists (Supplementary Fig. 5a, b). The antagonist is close to TM7, and the methoxycarbonyl group sterically prevents $K294^{7.36}$ from accessing the polar head (Fig. 4c and Supplementary Fig. 5a, b). By contrast, the agonist is closer to TM3 than the antagonist. Since the agonist closely interacts with TM7, the extracellular side of TM7 is shifted inwardly by 1.3 Å (Fig. 4c–e). Accompanied by the shift of TM7, $A300^{7.42}$, and $N303^{7.45}$ move toward TM6 and push the $W271^{6.48}$ rotamers inwardly (Fig. 4d, f). $W271^{6.48}$ is a part of the $C^{6.47}W^{6.48}xP^{6.50}$ motif, an essential mechanical activation switch conserved in class A GPCRs[30].

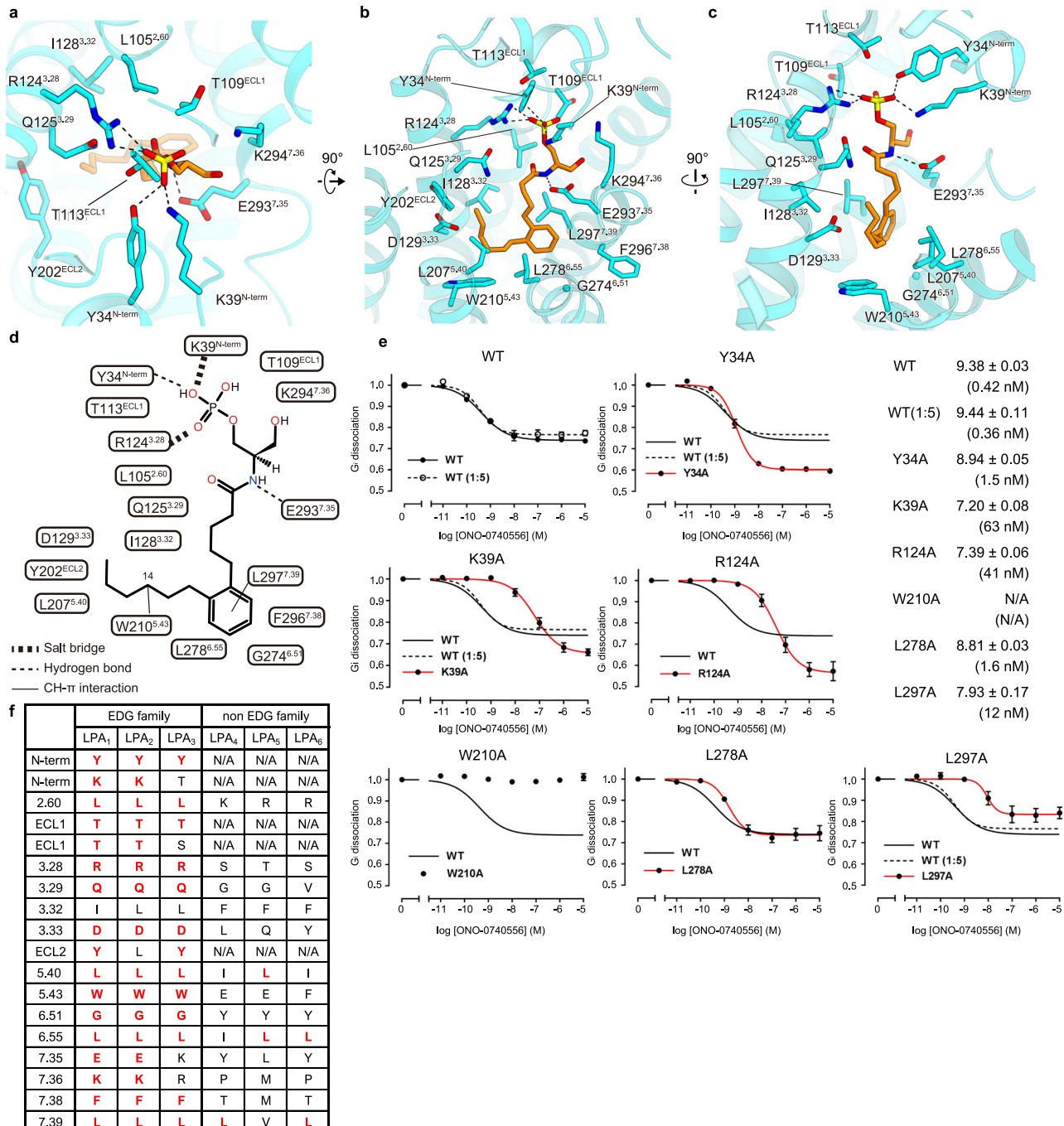

**Fig. 2 | ONO-0740556 binding site.** Binding pocket for ONO-0740556, viewed from the extracellular side (**a**) and membrane plane (**b**, **c**). ONO-0740556 and receptor residues involved in agonist binding are shown as orange and blue sticks, respectively. The dashed lines indicate hydrogen bonds. **d** Schematic representation of the interactions between ONO-0740556 and the receptor within 4.5 Å. **e** NanoBiT-G-protein dissociation assays for LPA₁ and its mutants. Concentration–response curves for ONO-074055-dependent G-protein dissociation signals for LPA₁ are shown as means ± s.e.m. from three independent experiments. To match the expression of LPA₁-WT to those of mutants with lower expression, 1:5 volume [WT (1:5)] plasmid was used. Source data are provided as a Source Data file. It should be noted that mutations of these head phosphate group-recognizing residues enhanced the G-protein dissociation in high concentration of the agonist, while the reason is uncertain. **f** Conservation of the ONO-0740556 binding site in LPA₁₋₆.

These observations suggest that the agonist interaction with TM7 affects the essential residue W271$^{6.48}$, leading to the receptor activation on the intracellular side, as discussed later.

The bottom of the pocket also affects the rearrangement of the C$^{6.47}$W$^{6.48}$xP$^{6.50}$ motif. In the antagonist-bound structure[24], L132$^{3.36}$, W210$^{5.43}$, and W271$^{6.48}$ constitute the bottom of the pocket, forming extensive hydrophobic interactions with the antagonist. Notably, L132$^{3.36}$ forms a CH–π interaction with W271$^{6.48}$, stabilizing the inactive conformation, while in the agonist-bound structure, C14 in the acyl chain of the agonist forms CH–π interactions with W210$^{5.43}$ and induces its side chain flipping. The rotamer change of W210$^{5.43}$ leads L132$^{3.36}$ to point towards the ligand. These structural changes weaken the interaction between L132$^{3.36}$ and W271$^{6.48}$ and allow their synergistic conformational changes (Fig. 4f and Supplementary Fig. 5c). A similar structural rearrangement is observed in CB₁, in which the homologous residues F200$^{3.36}$ and W356$^{6.48}$ are flipped

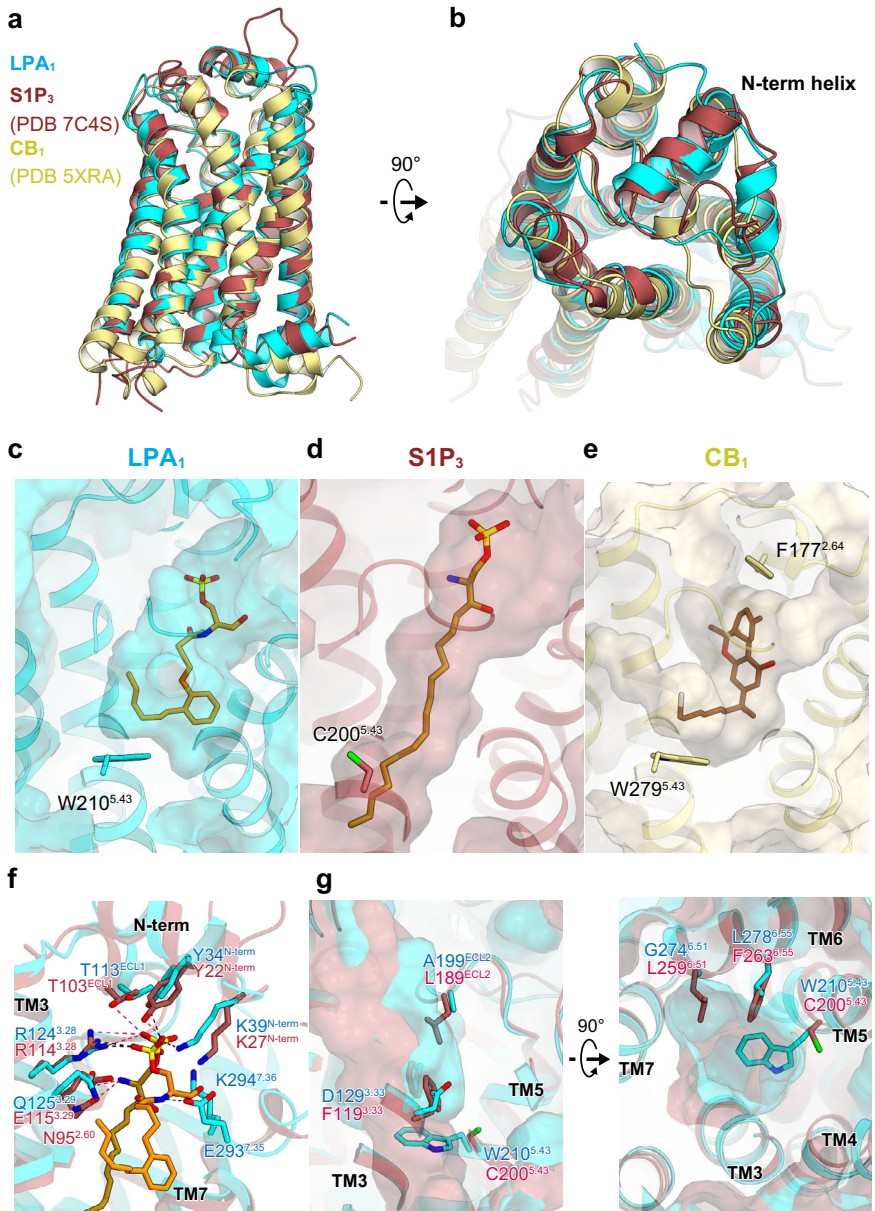

**Fig. 3 | Comparison of lipid binding modes. a, b** Superimposition of the agonist-bound LPA$_1$, S1P$_3$ (PDB 7C4S), and CB$_1$ (PDB 5XRA) structures, viewed from the membrane plane (**a**) and the extracellular side (**b**). Cross sections of the ligand binding pockets in LPA$_1$ (**c**), S1P$_3$ (**d**), and CB$_1$ (**e**). Superimposition of the LPA$_1$ and S1P$_3$ structures, focused on the phosphate recognition site (**f**) and the bottom of the hydrophobic pocket (**g**).

upon agonist binding (referred to as a twin toggle switch)[18,19,28] (Supplementary Fig. 5d). The density corresponding to C14 in the agonist is relatively well-observed (Fig. 1d), and the W210$^{5.43}$A mutant showed no G$_i$ dissociation signal (Fig. 2e), indicating the strength and importance of the interaction with W210$^{5.43}$. These observations suggest that the inward movement of TM7 and the acyl chain interaction with W210$^{5.43}$ cooperatively induce the toggle switch activation of W271$^{6.48}$ (Fig. 4f).

The movement of the C$^{6.47}$W$^{6.48}$xP$^{6.50}$ motif upon agonist binding causes a structural rearrangement in the P$^{5.50}$I$^{3.36}$F$^{6.44}$ motif, which is also essential for receptor activation[30–32]. The inward rotations of the W271$^{6.48}$ rotamer and N303$^{7.45}$ allow the F267$^{6.44}$ flipping toward TM5 (Fig. 4f), followed by the significant displacement of F218$^{5.51}$ proximal to the motif (Fig. 4g). The movement of the P$^{5.50}$I$^{3.40}$F$^{6.44}$ motif is responsible for the large outward movement of the intracellular portion of TM6. Accompanying the movement, structural rearrangements

are observed in the N$^{7.49}$P$^{7.50}$xxY$^{7.53}$ and D$^{3.49}$R$^{3.50}$Y$^{3.51}$ motifs conserved in most class A GPCRs[30]. In the N$^{7.49}$P$^{7.50}$xxY$^{7.53}$ motif, Y311$^{7.53}$ shows a significant displacement toward the intracellular core and contacts L139$^{3.43}$, I142$^{3.46}$, and R146$^{3.50}$, leading to the inward movement of TM7 (Fig. 4h). In the D$^{3.49}$R$^{3.50}$Y$^{3.51}$ motif, R146$^{3.50}$ forms a hydrogen bond with Y225$^{5.58}$ and enables interactions with the C-terminal residues of the α5-helix of G$_i$ (Fig. 4I).

These conformation changes create an intracellular cavity for G-protein recognition (Fig. 5a). The cavity closely contacts with the C-terminal α5-helix of G$_i$, which is the primary determinant for the G-protein coupling[25]. Specifically, R146$^{3.50}$ forms a hydrogen bond with the backbone carbonyl of C351$^{G.H5.23}$ (superscript indicates the common Gα numbering [CGN] system[33]), which is typically observed in other GPCR-G$_i$ complexes[34,35]. Additional hydrogen-bonding interactions are observed between the α5-helix and ICL2 (Fig. 5a). In addition to these polar contacts, there are extensive hydrophobic contacts

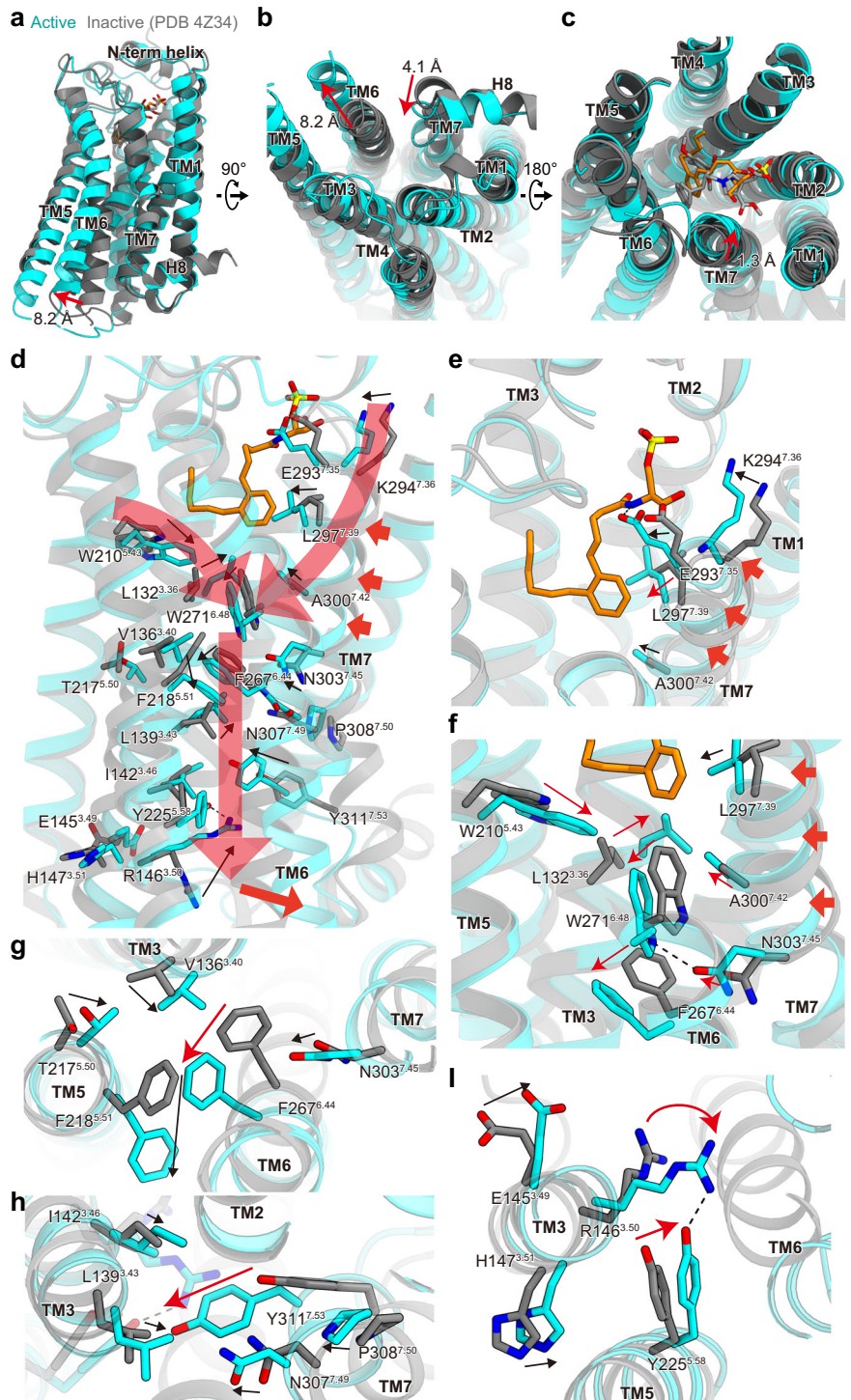

**Fig. 4 | Comparison of the agonist- and antagonist-bound LPA₁ structures.** Superimposition of the agonist- and antagonist-bound LPA₁ structures, colored cyan and gray (PDB 4Z34), respectively, viewed from the membrane plane (**a**), the intracellular side (**b**), and the extracellular side (**c**). D251 in TM6 and Y311 in TM7 are shifted by about 8.2 and 4.1 Å, respectively. **d–f** Structural changes of the intramolecular interactions induced by agonist binding (**d**). Panels (**e**) and (**f**) are focused on the extracellular side of TM7 and the receptor core, respectively. Rearrangement of the PIF (**g**), NPxxY (**h**), and DRY (**I**) motifs upon receptor activation. Hydrogen bonds are indicated by black dashed lines.

between the receptor and G$_i$ (Supplementary Fig. 6). These interactions allow the receptor to couple with G$_i$.

## Structural polymorphism at the receptor-G$_i$ interface

Previous structural studies showed that the Gα$_i$ binding manner is variable, with different Gα$_i$ rotations relative to the receptor[35]. Moreover, canonical (C) and non-canonical (NC) states were observed in the NTSR1-G$_i$ complex, with a 45° rotation of the G-protein relative to the receptor[36]. Compared with the C and NC states, the G$_i$ protein in the LPA₁ structure resides in their intermediate positions (Fig. 5b, c). This difference seems to be derived from the receptor-G$_i$ interaction at ICL2. In most class A GPCRs, ICL2 adopts a short α-helix in the active state[18,19,37–39]. Position$^{ICL2/34.50}$ (F174 in the NTSR1 C state) binds within the hydrophobic pocket formed by L194$^{G.S3.01}$, F336$^{G.H5.08}$, T340$^{G.H5.12}$,

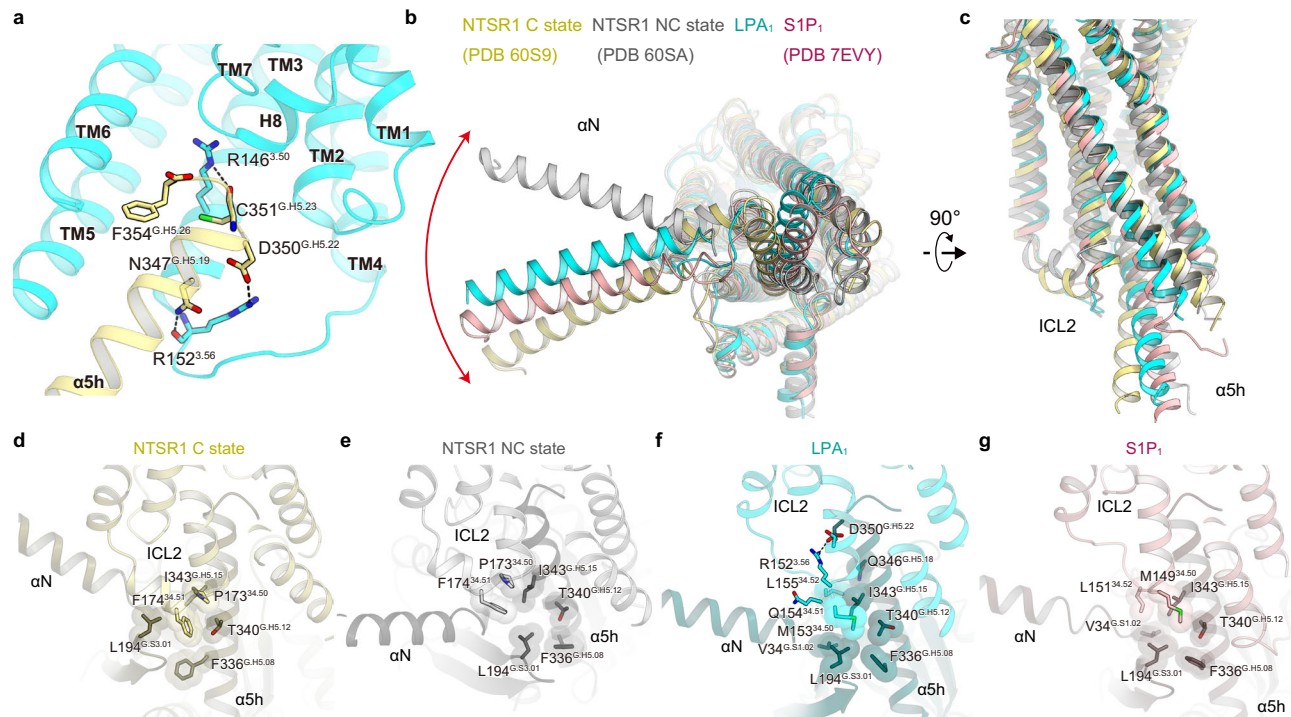

**Fig. 5 | Binding modes of Gᵢ. a** Main hydrogen-bonding interactions between the receptor and the α5 helix of Gαᵢ₁. **b, c** Structural comparisons of LPA₁-Gᵢ with other GPCR-G-protein complexes at the interface, viewed from the cytoplasmic side (**b**) and membrane plane (**c**). Structural comparisons of the interactions between ICL2 and Gᵢ in the NTSR1 C state (**d**), NTSR1 NC state (**e**), LPA₁ (**f**), and S1P₁ (**g**). Residues are shown as stick models. Hydrogen bonds are indicated by black dashed lines.

and I343^G.H5.15 in Gαᵢ (Fig. 5d) and plays an essential role in G-protein activation. In the NTSR1 NC state, F174^ICL2/34.51 is located away from the hydrophobic pocket of Gᵢ (Fig. 5e), closely correlating with the Gᵢ position (Fig. 5b, c). By contrast, the ICL2 of LPA₁ adopts a disordered conformation (Fig. 5f). The corresponding residue at position ICL2/ 34.51 is glutamine in LPA₁, which does not participate in a hydrophobic interaction. Instead, M153^ICL2/34.50 binds within the hydrophobic pocket. Moreover, L155^34.52 forms hydrophobic interactions with L343^G.H5.15 and Q346^G.H5.18. Above these hydrophobic interactions, R152^3.56 forms a salt bridge with D350^G.H5.22. Taken together, the disordered ICL2 tightly interacts with the α5-helix, shifting it away from ICL2 and TM3 as compared with the NTSR1 C state (Fig. 5c). These structural features are responsible for the different positions of the Gᵢ protein in the NTSR1 C and NC states. The disordered ICL2 and the G-protein position in the LPA₁-Gᵢ complex are similar to those in other S1P-Gᵢ complexes[21–23] (Fig. 5b, c, g), illuminating the conserved structural feature for Gᵢ coupling in EDG family members.

To determine whether the conformational transition of the Gᵢ coupling is observed, as in NTSR1, we performed 3D classifications focusing on the alignment of LPA₁ and G protein. Accordingly, we obtained cryo-EM maps for four classes (S1–S4) with nominal resolutions of 3.7, 3.9, 4.5, and 5.6 Å (PDB 7YU5, 7YU6, 7YU7, and 7YU8) (Fig. 6a and Supplementary Table 2). The maps of S1 and S2 enabled model building and refinement. Moreover, those of S3 and S4 enabled them with accuracy of the Cα atoms (Supplementary Fig. 7). Thus, we discuss the conformational changes in the main chains.

To visualize the G-protein movement, we superimposed the 3.5 Å-resolution structure described above (stable state) and S1–4 at the receptor. S1 and S2 superimposed well on the stable state, with limited in-plane rotations within 3–4° of the G protein (Fig. 6b), suggesting that this is the most stable position of the G-protein in the nucleotide-free state. By contrast, S3 and S4 both moved from the stable state in opposite directions from each other (Fig. 6c). As compared to S3, the entire Gαᵢ₁ in S4 is shifted downward by about 4 Å, followed by the

lateral movement of Gβ₁ by 5.3 Å. Focusing on the α5-helix, it moves 3 Å away from the receptor with the structural changes in the C-terminal residues (Fig. 6d). ICL2 follows the movement of the α5-helix to maintain the interaction with it. When aligned the S1–S4 and the stable state at the Gαᵢ₁ protein, the orientations of the C-terminal 2-turn helix are variable (Fig. 6e). This region does not adopt the α-helix in the GDP-bound inactive Gᵢ heterotrimer[40], and receptor interaction induces its helix formation. This notion suggests the innate structural flexibility in the C-terminal residues of the α5-helix, which is responsible for the structural polymorphism observed in this study, reflecting the dynamic equilibrium of the receptor-Gᵢ interface. These movements are totally distinct from the rotational movements observed in NTSR1 (Fig. 5b), which reflect the activation pathway of G protein[36]. Since the downward movement of Gαᵢ₁ weakens the receptor–Gαᵢ₁ interactions (Fig. 6d), S4 might represent the dissociation process of the receptor and G protein upon GTP binding.

## Discussion

We determined the structure of the LPA₁-Gᵢ complex bound to the LPA analog ONO-0740556, which revealed the tight recognition of the head phosphate and the accommodation of the bent acyl chain in the spherical pocket. Close examination of the active and inactive LPA₁ structures elucidated that two factors cooperatively play key roles in receptor activation. One is the recognition of the phosphate groups and glycerol backbone by TM7, and the other is the hydrophobic interactions with a long acyl chain by the residues at the bottom of the pocket. The ligand recognition by TM7 agrees with the properties of the binding module, in which the ligand is closer to TM7 and the hydrophobic pocket is more expanded to TM7 in LPA₁ than in S1P₃ (Fig. 3f, g). This is unique to LPA₁, among the currently reported structures of lysophospholipid receptors. At the binding site, the antagonist impedes receptor activation by its methoxycarbonyl group and indan, which prevent the inward movement of TM7. Moreover, at the bottom of the pocket, the position of the dimethoxyphenyl clashes

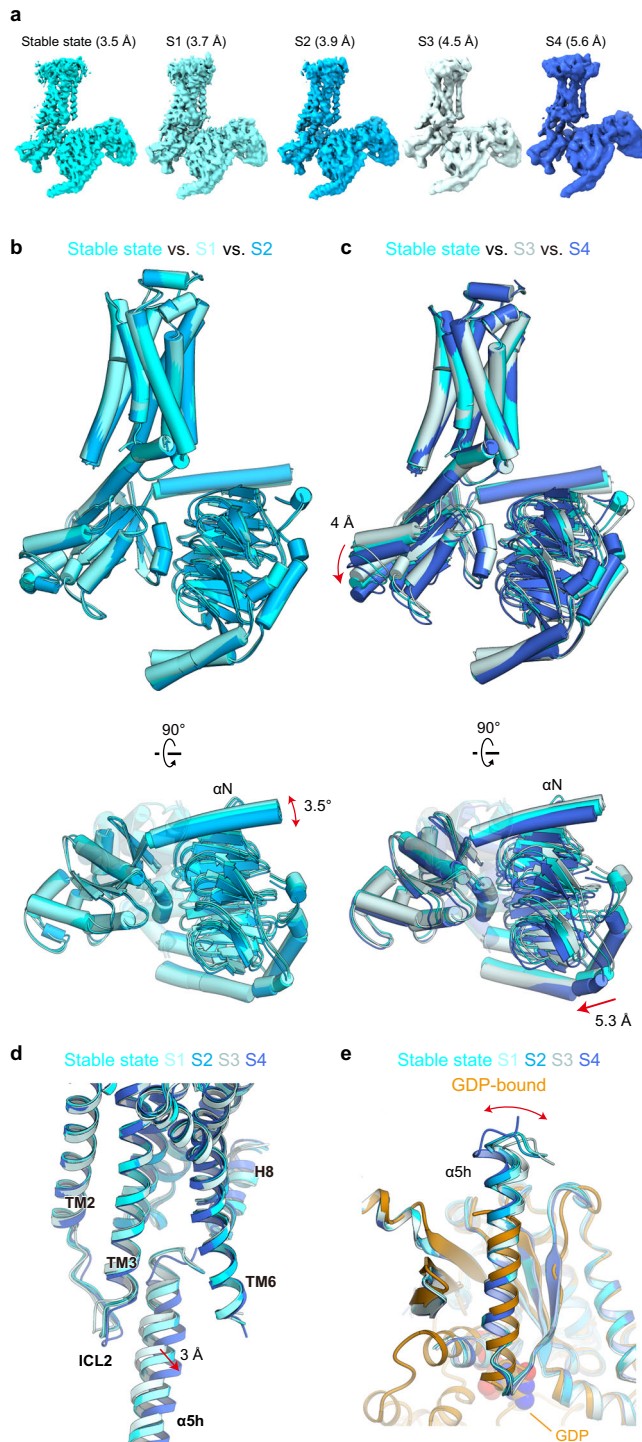

**Fig. 6 | Structural polymorphism of the LPA₁-G$_i$ interface. a** Density maps of the 3.5 Å resolution stable state and S1–4. **b** Comparison of the stable state, S1, and S2. **c** Comparison of the stable state, S3, and S4. **d** Superimposition of the stable state and S1–4, aligned at the receptor. TM5 is omitted. **e** Superimposition of the Gα$_{i1}$ subunits in the stable state, S1–4, and the GDP-bound inactive G$_i$ heterotrimer (PDB 1GG2).

with L132$^{3.36}$ and L297$^{7.39}$, which are essential for receptor activation (Supplementary Fig. 5a, e). This observation indicates that the antagonist inhibits L132$^{3.36}$ and L297$^{7.39}$ from moving towards activation.

After the submission of this manuscript, the structure of LPA₁ bound to the endogenous agonist LPA was reported (PDB 7TD0)[41]. We

performed a structural comparison of the LPA₁ bound to LPA with our structure bound to ONO-0740556. The two structures of LPA₁ superimposed well with a root mean square deviation of Cα atoms of 0.583 Å (Fig. 7a), and there are no significant differences in the recognition of polar regions on the extracellular side, and in the interaction of the ligand with W210$^{5.43}$ and L297$^{7.39}$ in the hydrophobic pocket (Fig. 7b, c). Given that these interactions mediate the receptor activation, LPA and ONO-0740556 activate the receptor in similar manners. However, interestingly, the route of the acyl chain is different between our new compound and LPA. The acyl chain of LPA folds on the TM5 side and extends toward TM7, but ONO-0740556 goes from TM7 to TM5 (Fig. 7c). This fact suggests that LPA₁ permits the acceptance of various forms of acyl chains within the spherical hydrophobic pocket. The interaction of L297$^{7.39}$ with the hydrocarbon chain of LPA is weaker than the CH–π interaction with the aromatic moiety of ONO-0740556. This difference would be one of the factors causing the distinct affinities of the agonists (Fig. 1b and Supplementary Table 1). Our study clarifies the detailed structure-activity relationship of LPA₁ and will facilitate the design of novel LPA-mimetic agonists to explore the therapeutic potential of LPA₁.

G$_i$ movement was observed in the LPA-bound LPA₁-G$_i$ complex, as in our study. The 3D variability analysis (3DVA) of the LPA-bound complex identified two states (Fig. 7d) distinguished by the relative rotation of Gα$_{i1}$ about LPA₁ in the plane of the membrane, ~5° in both directions away from the consensus structure[41]. By contrast, in the ONO-0740556 bound complex, the entire Gα$_{i1}$ in S4 is shifted downward by about 4 Å (Fig. 7e), indicating a weakening of the receptor–Gα$_{i1}$ interactions. However, there are significant differences in the experimental conditions between our study and previous studies (e.g., ligands, detergents, analysis methods, etc.), and thus we cannot ignore their influence on the G$_i$ movements. Moreover, the 3DVA analysis of LPA₁ and S1P₁ elucidated the rocking, twisting, and flexing motions of the receptor about the G protein[41]. These structural polymorphisms indicated the flexible coupling between GPCR and G protein, which may be observed in other GPCR-G-protein complexes by more careful analysis. The G-protein movement upon dissociation is also observed in the recently reported PTH1R-G$_s$ complex[42]. Future studies will shed light on whether the observed structural polymorphism reflects the structural flexibility in the purified condition, or the process of G-protein activation and dissociation by GPCRs.

## Methods

### NanoBiT-G-protein dissociation assay

LPA₁-induced G$_i$ activation was measured by a NanoBiT-G-protein dissociation assay[25], in which the LPA₁-induced dissociation of a Gα subunit from a Gβγ subunit was monitored by a NanoBiT system (Promega). Specifically, a NanoBiT-G$_{i1}$ protein consisting of a large fragment (LgBiT)-containing Gα$_{i1}$ subunit and a small fragment (SmBiT)-fused Gγ₂ subunit with the C68S mutation, along with the untagged Gβ₁ subunit, was expressed with a test LPA₁ construct, and the ligand-induced change in the luminescent signal was measured. We used the N-terminal FLAG (DYKDDDDK)-tagged constructs of human LPA₁. HEK293T cells were seeded in a six-well culture plate at a concentration of $2 \times 10^5$ cells ml$^{-1}$ (2 ml per well in DMEM supplemented with 10% fetal bovine serum), 1 d before transfection. The transfection solution was prepared by combining 2.5 μl (per well hereafter) of Lipofectamine 2000 (ThermoFisher Scientific) and a plasmid mixture consisting of 100 ng LgBiT-containing Gα$_{i1}$ subunit, 500 ng Gβ₁, 500 ng SmBiT-fused Gγ₂ with the C68S mutation, and 200 ng LPA₁ in 500 μl of Opti-MEM (ThermoFisher Scientific). After an incubation for 1 d, the transfected cells were harvested with 0.5 mM EDTA-containing PBS, centrifuged, and suspended in 2 ml of HBSS containing 0.01% bovine serum albumin (BSA fatty acid-free grade, SERVA) (assay buffer). The cell suspension was dispensed into a white 96-well plate at a volume of 80 μl per well, and loaded with 20 μl of 50 μM coelenterazine diluted in

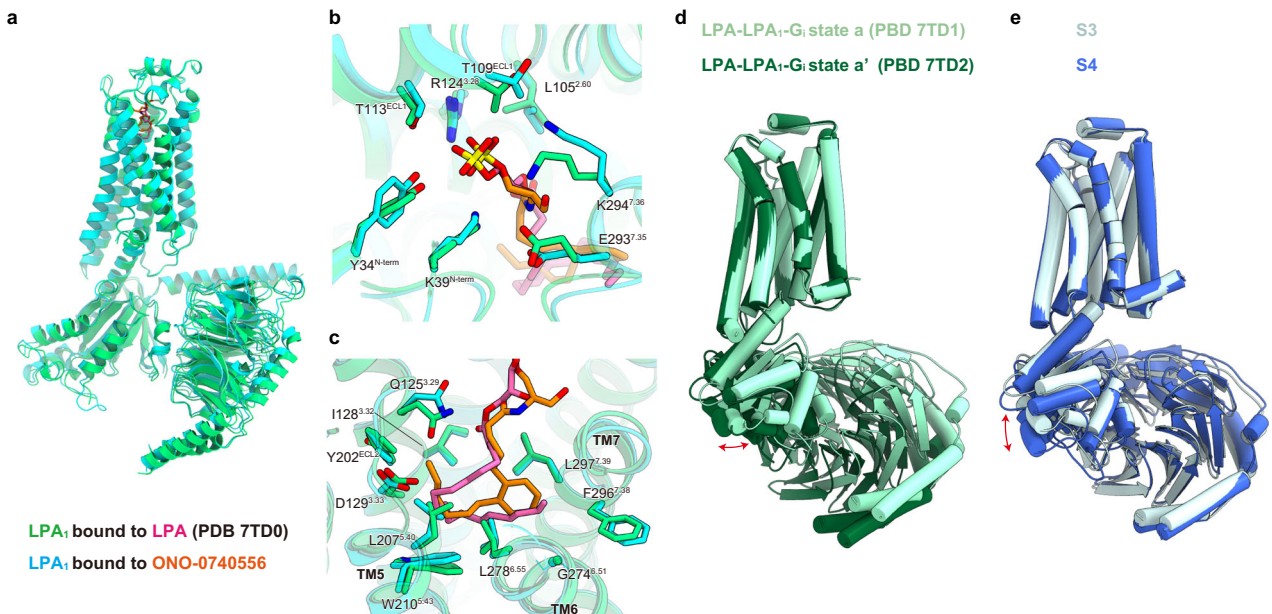

**Fig. 7 | Structural comparison of LPA$_1$ bound to LPA and ONO-0740556.**
**a** Superimposition of the LPA- and ONO-0740556-bound LPA$_1$ structures, colored green (PDB 7TD0) and cyan, respectively. **b**, **c** Superimposition of the binding pocket for LPA and ONO-0740556 in polar regions on the extracellular side (**b**), and in the hydrophobic pocket (**c**). **d** Superimposition of LPA-LPA$_1$-G$_i$ states a (PDB 7TD1) and a' (PDB 7TD2) aligned at the receptor. **e** Superimposition of S3 and S4 aligned at the receptor.

the assay buffer. After 2 h of incubation, the plate was measured for baseline luminescence and then 20 μl of 6× test compound, diluted in the assay buffer, was manually added. After an incubation for 8–10 min at room temperature, the plate was read for the second measurement. The second luminescence counts were normalized to the initial counts, and the fold changes in the signals were plotted for the G-protein dissociation response. Using the Prism 7 software (GraphPad Prism), the G-protein dissociation signals were fitted to a four-parameter sigmoidal concentration–response curve, from which the pEC$_{50}$ values (negative logarithmic values of half-maximum effective concentration (EC$_{50}$) values) and Emax were used to calculate the mean and s.e.m.

### Measurement of receptor cell-surface expression by ELISA
To measure the cell surface expression level of wild-type LPA$_1$ and its mutants, HEK293T cells were transiently transfected in 12-well plates and incubated for 1 d. Transfection was performed by following the same procedure as described in the NanoBiT-G-protein dissociation assay section, with a downscaled volume (250 μl transfection solution). The transfected cells were harvested with 0.5 mM EDTA-containing PBS, centrifuged, and blocked with 5% (w/v) BSA at room temperature for 1 h. Anti-Flag HRP conjugate (Sigma) was then added to a dilution of 1:20,000 and incubated for 1 h at room temperature. After washing with PBS, the cells were suspended in 100 μL of PBS and 10 μl portions were dispensed into the tubes. A 50 μl aliquot of TMB, HRP Microwell Substrate (SurModics, Inc.) was added. The reactions were quenched by adding an equal volume of 450 nm Liquid Stop Solution for TMB Microwell Substrate (SurModics, Inc.) and the optical density at 450 nm was measured using Nanodrop One (Thermo Fischer Scientific).

### Expression and purification of the human LPA$_1$
The human LPA$_1$ gene (UniProtKB, Q92633) was subcloned into a modified pFastBac vector[43], with an N-terminal haemagglutinin signal peptide followed by the Flag-tag epitope (DYKDDDD) and a C-terminal tobacco etch virus (TEV) protease recognition site followed by an EGFP-His[10] tag. The recombinant baculovirus was prepared using the Bac-to-Bac baculovirus expression system (Thermo Fischer Scientific).

*Spodoptera frugiperda* Sf9 insect cells (Thermo Fischer Scientific) were infected with the virus at a cell density of $4.0 \times 10^6$ cells per milliliter in Sf900 II medium (Gibco), and grown for 48 h at 27 °C. The harvested cells were disrupted by sonication, in buffer containing 20 mM Tris-HCl, pH 8.0, 200 mM NaCl, and 10% glycerol. The crude membrane fraction was collected by ultracentrifugation at $180,000 \times g$ for 2 h. The membrane fraction was solubilized in buffer, containing 20 mM Tris-HCl, pH 8.0, 150 mM NaCl, 1% n-dodecyl-beta-D-maltopyranoside (DDM) (Calbiochem), 0.2 % CHS, 10% glycerol, and 2 μM ONO-0740556 for 1 h at 4 °C. The supernatant was separated from the insoluble material by ultracentrifugation at $180,000 \times g$ for 30 min, and incubated with TALON resin (Clontech) for 30 min. The resin was washed with ten column volumes of buffer, containing 20 mM Tris-HCl, pH 8.0 500 mM NaCl, 0.05% glyco-diosgenin (GDN) (Anatrace), 0.1 μM ONO-0740556, and 15 mM imidazole. The receptor was eluted in buffer, containing 20 mM Tris-HCl, pH 8.0, 500 mM NaCl, 0.01% GDN, 0.1 μM ONO-0740556, and 200 mM imidazole. The receptor was concentrated and loaded onto a Superdex200 10/300 Increase size-exclusion column, equilibrated in buffer containing 20 mM Tris-HCl, pH 8.0, 150 mM NaCl, 0.01% GDN, and 0.1 μM ONO-0740556. Peak fractions were pooled and frozen in liquid nitrogen.

### Expression and purification of the G$_i$ heterotrimer
The G$_i$ heterotrimer was expressed and purified using the Bac-to-Bac baculovirus expression system, according to the method reported previously[35]. Sf9 insect cells were infected at a density of $3–4 \times 10^6$ cells ml$^{-1}$ with a one 100th volume of two viruses, one encoding the WT human Gα$_{i1}$ subunit and the other encoding the WT bovine Gγ$_2$ subunit and the WT rat Gβ$_1$ subunit containing a His$_8$ tag followed by an N-terminal TEV protease cleavage site. The infected Sf9 cells were incubated in Sf900II medium at 27 °C for 48 h. The Sf9 cells were collected by centrifugation at $6200 \times g$ for 10 min. The collected cells were lysed in buffer containing 20 mM Tris, pH 8.0, 150 mM NaCl, and 10% glycerol. The Gα$_{i1}$β$_1$γ$_2$ heterotrimer was solubilized at 4 °C for 1 h, in buffer containing 20 mM Tris (pH 8.0), 150 mM NaCl, 10% glycerol, 1% (w/v) n-dodecyl-beta-D-maltopyranoside (DDM) (Anatrace), 50 μM GDP (Roche), and 10 mM imidazole. The soluble

fraction containing $G_i$ heterotrimers was isolated by ultra-centrifugation (186,000 × $g$ for 20 min) and the supernatant was mixed with Ni-NTA Superflow resin (Qiagen) and stirred at 4 °C for 1 h. The resin was washed with 10 column volumes of buffer, containing 20 mM Tris, pH 8.0, 150 mM NaCl, 0.02% DDM, 10% glycerol, 10 µM GDP, and 30 mM imidazole. Next, the $G_i$ heterotrimers were eluted with two column volumes of buffer, containing 20 mM Tris, pH 8.0, 150 mM NaCl, 0.02% (w/v) DDM, 10% (v/v) glycerol, 10 µM GDP and 300 mM imidazole. The eluted fraction was dialyzed overnight at 4 °C against 20 mM Tris, pH 8.0, 50 mM NaCl, 0.02% DDM, 10% glycerol, and 10 µM GDP. To cleave the histidine tag, TEV protease was added during the dialysis. The dialyzed fraction was incubated with Ni-NTA Superflow resin at 4 °C for 1 h. The flow-through was collected and purified by ion-exchange chromatography on a HiTrapQ HP column (GE), using buffer I1 (20 mM Tris, pH 8.0, 50 mM NaCl, 0.02% DDM, 10% glycerol, and 1 µM GDP) and buffer I2 (20 mM Tris, pH 8.0, 1 M NaCl, 0.02% DDM, 10% glycerol, and 1 µM GDP).

### Expression and purification of scFv16

The gene encoding scFv16 was synthesized (GeneArt) and subcloned into a modified pFastBac vector, with the resulting construct encoding the GP67 secretion signal sequence at the N terminus, and a $His_8$ tag followed by a TEV cleavage site at the C terminus[35]. The $His_8$-tagged scFv16 was expressed and secreted by Sf9 insect cells, as previously reported[35] The Sf9 cells were removed by centrifugation at 5000 × $g$ for 10 min, and the secreta-containing supernatant was combined with 5 mM $CaCl_2$, 1 mM $NiCl_2$, 20 mM HEPES (pH 8.0), and 150 mM NaCl. The supernatant was mixed with Ni Superflow resin (GE Healthcare Life Sciences) and stirred for 1 h at 4 °C. The collected resin was washed with buffer containing 20 mM HEPES (pH 8.0), 500 mM NaCl and 20 mM imidazole, and further washed with 10 column volumes of buffer containing 20 mM HEPES (pH 8.0), 500 mM NaCl and 20 mM imidazole. Next, the protein was eluted with 20 mM Tris (pH 8.0), 500 mM NaCl and 400 mM imidazole. The eluted fraction was concentrated and loaded onto a Superdex200 10/300 Increase size-exclusion column, equilibrated in buffer containing 20 mM Tris (pH 8.0) and 150 mM NaCl. Peak fractions were pooled, concentrated to 5 mg ml$^{-1}$ using a centrifugal filter device (Millipore 10 kDa MW cutoff), and frozen in liquid nitrogen.

### Formation and purification of the LPA$_1$-G$_i$ complex

Purified LPA$_1$-GFP was mixed with a 1.2 molar excess of $G_i$ heterotrimer, ScFv16, and TEV protease. After the addition of apyrase to catalyze hydrolysis of unbound GDP, and ONO-0740556 (final 10 µM) the coupling reaction was performed at 4 °C for overnight. To remove excess G protein, the complexing mixture was purified by M1 anti-Flag affinity chromatography. Bound complex was washed in buffer, containing 20 mM Tris-HCl, pH8.0, 150 mM NaCl, 0.01% GDN, 10 µM ONO-0740556, 10% Glycerol, and 5 mM $CaCl_2$. The complex was then eluted in 20 mM Tris-HCl, pH8.0, 150 mM NaCl, 0.01% GDN, 10 µM ONO-0740556, 10% Glycerol, 5 mM EDTA, and Flag peptide. The LPA$_1$-G$_i$-scFv16 complex was purified by size exclusion chromatography on a Superdex 200 10/300 column in 20 mM Tris-HCl, pH8.0, 150 mM NaCl, 0.01% GDN, and 1 µM ONO-0740556. Peak fractions were concentrated to ~12 mg/ml for electron microscopy studies.

### Sample vitrification and cryo-EM data acquisition

The purified complex was applied onto a freshly glow-discharged Quantifoil holey carbon grid (R1.2/1.3, Au, 300 mesh), and plunge-frozen in liquid ethane by using a Vitrobot Mark IV. Data collections were performed on a 300 kV Titan Krios G3i microscope (Thermo Fisher Scientific) and equipped with a BioQuantum K3 imaging filter and a K3 direct electron detector (Gatan). In total, 6,227 movies were acquired with a calibrated pixel size of 0.83 Å pix$^{-1}$ and with a defocus range of −0.8 to −1.6 µm, using the SerialEM software[44]. Each movie was acquired for 2.57 s and split into 48 frames, resulting in an accumulated exposure of about 49.530 e$^-$ Å$^{-2}$ at the grid.

### Image processing

All acquired movies were dose-fractionated and subjected to beam-induced motion correction implemented in RELION 3.1[45]. The contrast transfer function (CTF) parameters were estimated using CTFFIND 4.0[46] (Rohou & Grigorieff, 2015). A total of 3,021,676 particles were extracted. The initial model was generated in RELION 3.1[47,48]. The particles were subjected to several rounds of 2D and 3D classifications, resulting in the optimal classes of particles, which contained 363,784 particles. Next, the particles were subjected to 3D refinement, CTF refinement, and Bayesian polishing[49] (Zivanov et al., 2018).

The GDN detergent micelles and the α-helical domain of the $G\alpha_i$ subunit of the 363,784 particles were subtracted to obtain a higher signal-to-noise ratio. The subtracted particles were subjected to 3D classifications. The best class of particles was subjected to 3D refinement and then subjected to No-alignment classifications. The best class of particles were subjected to 3D refinement, postprocessing yielded a map with a nominal overall resolution of 3.5 Å, with the gold standard Fourier Shell Correlation (FSC = 0.143) criteria[50].

Moreover, the 3D model was refined with a mask on the receptor. As a result, the receptor has a higher resolution with a nominal resolution of 3.7 Å. The local resolution was estimated by RELION 3.1. The processing strategy is described in Fig. S2.

Apart from that, the 3D model of the 363,784 particles was refined with a mask on the $G_i$ and ScFv16, and then No-alignment classification with a mask on receptor obtained multiple conformations with different $G_i$-couplings. Each class of particles was subjected to 3D refinement, micelles and α-helical domains were subtracted and then was subjected to 3D refinement.

### Model building and refinement

The quality of the micelle-subtracted density map was sufficient to build a model manually in COOT[51,52]. The model building was facilitated by the predicted LPA$_1$ model in AlphaFold Protein Structure Database (https://alphafold.ebi.ac.uk/entry/Q92633) and the cryo-EM structure of the LPA$_1$-G$_i$ and µOR-G$_i$ complex (PDB 7TD0 and 6DDE, respectively)[34,41]. We manually modeled LPA$_1$, the $G_i$ heterotrimer and scFv16 into the map by jiggle fit using COOT. We then manually readjusted the model into the density map using COOT and refined it using phenix.real_space_refine[53,54] (v.1.19) with the secondary-structure restraints using phenix secondary_structure_restraints. Finally, we refined the model using servalcat[55].

### Reporting summary

Further information on research design is available in the Nature Research Reporting Summary linked to this article.

## Data availability

The data that support this study are available from the corresponding authors upon reasonable request. The cryo-EM density map and atomic coordinates for the LPA$_1$-G$_i$ complex have been deposited in the Electron Microscopy Data Bank and the PDB, under accession codes: EMD-34097 (LPA$_1$–G$_i$ stable state), EMD-34098 (focused on LPA$_1$), EMD-34099 (LPA$_1$–G$_i$ state 1), EMD-34100 (LPA$_1$–G$_i$ state 2), EMD-34101 (LPA$_1$–G$_i$ state 3), EMD-34102 (LPA$_1$–G$_i$ state 4), and PDB 7YU3 (LPA$_1$–G$_i$ stable state), 7YU4 (focused on LPA$_1$), 7YU5 (LPA$_1$–G$_i$ state 1), 7YU6 (LPA$_1$–G$_i$ state 2), 7YU7 (LPA$_1$–G$_i$ state 3), 7YU8 (LPA$_1$–G$_i$ state 4). Source data are provided with this paper.

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

## Acknowledgements

We thank K. Ogomori and C. Harada for technical assistance, Inoue, A for the advice on the assay experiment and Ono Pharmaceutical Co., Ltd. for the synthesis and characterization of ONO-0740556. This work was supported by grants from the Platform for Drug Discovery, Informatics and Structural Life Science by the Ministry of Education, Culture, Sports, Science and Technology (MEXT), and JSPS KAKENHI grants 21H05037 (O.N.), 22K19371 and 22H02751 (W.S.), and 21J20692 (T.T.); ONO Medical Research Foundation (W.S.); The Kao Foundation for Arts and Sciences (W.S.); The Takeda Science Foundation (W.S.); The Uehara Memorial Foundation (W.S.); the Platform Project for Supporting Drug Discovery and Life Science Research (Basis for Supporting Innovative Drug Discovery and Life Science Research (BINDS)) from AMED, under grant numbers JP19am01011115 (support no. 1109, O.N.).

## Author contributions

H.A. performed all of the experiments. T.T. assisted with the grid preparation, the cryo-EM data collection, and the single particle analysis. F.S. assisted with the single particle analysis. Y.M. assisted with the NanoBiT-G-protein dissociation assay. W.S. performed the initial screening of the $LPA_1$ expression. The manuscript was mainly prepared by H.A. and W.S., with assistance from O.N.

## Competing interests

O.N. is a co-founder and scientific advisor for Curreio. All other authors declare no competing interests.
