## [Peer Review File · Nature Communications]

REVIEWER COMMENTS

Reviewer #1 (Remarks to the Author):

This manuscript by Akasaka et al., describes the cryo-EM structure of LPA1R-Gi complex where the receptor is occupied with an LPA analog agonist. It reveals the binding mechanism of this LPA analog, which exhibits similarities with S1P receptors, and also provides potential binding mode of LPA to this receptor. A closer analysis of the agonist-binding pocket and comparison with other lipid-binding GPCRs uncovers interesting insights into the selectivity of LPA for LPA1R, and allows the rationalization of previous biochemical and computational studies. The receptor displays an activated conformation with outward movement of TM6, the active conformation of the structural motifs as observed in other activated GPCR structures. Interestingly, the Gi conformation is different from several of the other structures determined previously, a position that is somewhat intermediate between the canonical and non-canonical states as seen for NTS1R-Gi structures in the nanodisc. The differential involvement and structural element in ICL2 in the current structure is also quite intriguing and provides an important observation for future studies. There are a couple of other structures reported recently (e.g. Liu et al., 2022), however, the current study provides important novel information and significant advance over the previous structures, and therefore, the impact and novelty of the current study is not compromised by the previous publications. In fact, the current study, taken together with the earlier structures, now offers comprehensive structural templates to better understand the activation mechanism of lipid-sensing GPCRs and to employ these structural platforms for discovery of novel therapeutic agents. Therefore, I strongly recommend the publication of this manuscript in Nature Communications. I have only a few minor suggestions that the authors may consider while revising their manuscript.

1. As mentioned above, authors should consider a comparison and discussion of their findings in reference to previous structure by Liu et al., where the receptor is bound to LPA. Again, the previous structures do not compromise the novelty or impact of the current study but rather provides important information, and therefore, complementary in nature.

2. As the authors are using a newly developed analog of LPA, it may be worth characterizing it in a little more details in terms of pharmacology. Currently, the authors have included only the Ca⁺⁺ data but they may consider also testing its potency and efficacy, for example, in arrestin-recruitment assays. Again, this is not an absolutely requirement for the publication of this paper but may strengthen it further.

Arun K. Shukla, Ph.D.

Reviewer #2 (Remarks to the Author):

The authors present a valuable cryo-EM structure of the LPA1 receptor in complex with an agonist. Comparisons with active-state structures of other lipid receptors bound with agonists highlight differences in ligand recognition, and extensive analysis of polymorphisms in the receptor-Gi interface are also of interest. There are three areas in which the manuscript can be improved with regards to clarity of communication, one scientific concern that the authors should consider, and one minor error in need of correction.

Clarity of communication recommendations:

1. “Druggable” is an adjective generally applied to biomolecular targets that drugs might interact with to produce a positive therapeutic effect. The authors pair this adjective with “agonists”, which is not standard in the field. I recommend pairing “agonists” with a different adjective more commonly paired with small molecule ligands, such as “drug-like”.
2. The authors state that “CB1 lacks the N-term helix” in their comparison of CB1, S1P3 and LPA1 agonist bound structures. Given that only residues 104-112 of the CB N-terminus are resolved in PDB entry 5XRA, I believe this is not a supportable conclusion – had the entire N-terminus been present, it may or may not have displayed a comparable helical segment. The authors should acknowledge that most of the N-terminus of CB1 is absent instead of concluding that it does not display the N-terminal helix observed in LPA1 and S1P3.
3. The figure 6 legend indicates that structural polymorphism of the LPA1-Gi interface is presented, however the manuscript body indicates “To determine whether the conformational transition of the Gi coupling is observed in NTSR1, we performed 3D classification on the alignment of the receptor and the G protein” before referring to figure 6. The reference to two different receptors here is confusing. Which receptor was investigated for Gi coupling transitions?

Scientific concern:

1. The structure of the bound agonist (ONO-0740556) in the cryo-EM structure of the LPA1:agonist complex (Figure 1b) exhibits an unstable amide dihedral angle of -106.7 degrees (my appreciation to the authors for providing the structure to accompany this review so that the actual angle could be verified). Minimum energy conformations are expected to have amide dihedral angles near either 180 degrees or 0 degrees. Have the authors considered if a lower energy conformation of ONO-0740556 can be fitted into the cryo-EM map? If a more reasonable conformation cannot be fitted into the cryo_EM map,

discussion of this high energy conformation in the context of the complex should be added to the manuscript.

Minor errors to correct:

1. The structure of compound 5 in scheme 2 does not match the name of compound 5 defined in the supplemental methods. Compound 5 should be shown as the unprotected amine as the conditions defined for conversion of compound 5 to 6 should accomplish both the amine protection and the carboxylic acid reduction.

Signed: Abby L. Parrill

Reviewer #3 (Remarks to the Author):

This manuscript by Akasaka et al reports a cryo-EM structure of human LPA1-Gi complex bound to a LPA analog ONO-0740556. LPA1 participates in nervous-system tissue development and chondrocyte differentiation, as a drug target in the treatment of cancer, inflammation, and neuropathic pain. The authors provided structural insights into ligand binding, receptor activation, and G protein coupling of LPA1. Structural determinants for lipid preference among LPA1, S1P3, and CB1, and rearrangements of motifs during LPA1 activation are identified. The resolution is relatively low (3.5Å), but the density around the ligand could be observed. Generally, this manuscript is not well written to support the authors claims with several missing information in the text and figures. This reviewer suggests major revisions before publication. The authors are strongly encouraged to edit the manuscript extensively to improve the text and figures.

1. The affiliations for all authors are missing in the manuscript.
2. Line 188, the sentence "On the intracellular side, TM6 is displaced outward by about 11 Å, and TM7 is shifted inward by about 3.5 Å", it is better to point out which residues were used in the measurement.
3. Line 435, what does the "100th volume" refer to?
4. Figure 2a, all residues interacting with the ligand shown in sticks should be labeled.
5. Figure 2d, what are the red and blue boxes standing for?
6. Figure 2e and Figure S3, the authors are encouraged to provide the sequence alignment results, and color the residues to show if they are conserved or not.

7. Figure 4d, all residues are missing.
8. Table S1, the Ca²⁺ influx assay should be represented as Mean \pm S.E.M. from three independent experiments.
9. The authors are encouraged to perform the Ca²⁺ influx assay for the mutagenesis on the ligand-binding site to validate the receptor-ligand interactions identified in the structure.
10. Figure 1c, what are the different contours?
11. Figure S3 is a table and should be Table S3.
12. Figure S4, the structure of LPA1 antagonist should be added.
13. Figure S6, which helices are shown in the lower panel?
14. The title is "Structure and dynamics of Human Lysophosphatidic Acid Receptor 1", but no dynamics results are presented. The authors are encouraged to perform the molecular dynamics simulations to better support the author's claims.

Major changes

- We rebuilt the model of ONO-0740556 with the constraint that the amide dihedral angle is flat, according to the suggestion by Reviewer #2. There is no significant difference in the outline and only a few changes in the interactions with the residues. We rewrote some of the text and figures to reflect these changes.
- We performed the NanoBiT-G-protein dissociation assay instead of the Ca²⁺ influx assay to directly examine GPCR activation, because it is now difficult to perform the Ca²⁺ influx assay due to material and machinery problems. At this time, the Gi signal used in the structural analysis was measured. The mutational analysis suggested by Reviewer #3 was also performed with the NanoBiT-G-protein dissociation assay.

Reviewer #1

This manuscript by Akasaka et al., describes the cryo-EM structure of LPA1R-Gi complex where the receptor is occupied with an LPA analog agonist. It reveals the binding mechanism of this LPA analog, which exhibits similarities with SIP receptors, and also provides potential binding mode of LPA to this receptor. A closer analysis of the agonist-binding pocket and comparison with other lipid-binding GPCRs uncovers interesting insights into the selectivity of LPA for LPA1R, and allows the rationalization of previous biochemical and computational studies. The receptor displays an activated conformation with outward movement of TM6, the active conformation of the structural motifs as observed in other activated GPCR structures. Interestingly, the Gi conformation is different from several of the other structures determined previously, a position that is somewhat intermediate between the canonical and non-canonical states as seen for NTS1R-Gi structures in the nanodisc. The differential involvement and structural element in ICL2 in the current structure is also quite intriguing and provides an important observation for future studies. There are a couple of other structures reported recently (e.g. Liu et al., 2022), however, the current study provides important novel information and significant advance over the previous structures, and therefore, the impact and novelty of the current study is not compromised by the previous publications. In fact, the current study, taken together with the earlier structures, now offers comprehensive structural templates to better understand the activation mechanism of lipid-sensing GPCRs and to employ these structural platforms for discovery of novel therapeutic agents. Therefore, I strongly recommend the publication of this manuscript in Nature Communications. I have only a few minor suggestions that the authors may consider while revising their manuscript.

We appreciate the reviewer's favorable evaluation of our paper.

1. As mentioned above, authors should consider a comparison and discussion of their findings in reference

to previous structure by Liu et al., where the receptor is bound to LPA. Again, the previous structures do not compromise the novelty or impact of the current study but rather provides important information, and therefore, complementary in nature.

According to the suggestion, we added a comparison and discussion.

2. As the authors are using a newly developed analog of LPA, it may be worth characterizing it in a little more details in terms of pharmacology. Currently, the authors have included inly the Ca⁺⁺ data but they may consider also testing its potency and efficacy, for example, in arrestin-recruitment assays. Again, this is not an absolutely requirement for the publication of this paper but may strengthen it further.

According to the suggestion, we performed the NanoBiT-G-protein dissociation assay instead of the Ca²⁺ influx assay to directly examine GPCR activation (Fig. 1b, Table S1).

Reviewer #2

The authors present a valuable cryo-EM structure of the LPA1 receptor in complex with an agonist. Comparisons with active-state structures of other lipid receptors bound with agonists highlight differences in ligand recognition, and extensive analysis of polymorphisms in the receptor-Gi interface are also of interest. There are three areas in which the manuscript can be improved with regards to clarity of communication, one scientific concern that the authors should consider, and one minor error in need of correction.

We appreciate the reviewer's favorable evaluation of our paper.

Clarity of communication recommendations:

1. "Druggable" is an adjective generally applied to biomolecular targets that drugs might interact with to produce a positive therapeutic effect. The authors pair this adjective with "agonists", which is not standard in the field. I recommend pairing "agonists" with a different adjective more commonly paired with small molecule ligands, such as "drug-like".

According to the suggestion, we corrected them (lines 45, 87 and 102).

2. The authors state that "CBI lacks the N-term helix" in their comparison of CBI, SIP3 and LPA1 agonist bound structures. Given that only residues 104-112 of the CB N-terminus are resolved in PDB entry 5XRA, I

believe this is not a supportable conclusion – had the entire N-terminus been present, it may or may not have displayed a comparable helical segment. The authors should acknowledge that most of the N-terminus of CB1 is absent instead of concluding that it does not display the N-terminal helix observed in LPA1 and SIP3.

According to the suggestion, we corrected them, as follows:

CB1 lacks the N-term helix, exhibiting a difference.

→ the N-terminus of CB1 is different, with only partial structures observed (lines 154 and 155).

3. The figure 6 legend indicates that structural polymorphism of the LPA1-Gi interface is presented, however the manuscript body indicates “To determine whether the conformational transition of the Gi coupling is observed in NTSR1, we performed 3D classification on the alignment of the receptor and the G protein” before referring to figure 6. The reference to two different receptors here is confusing. Which receptor was investigated for Gi coupling transitions?

According to the suggestion, we corrected them, as follows:

To determine whether the conformational transition of the Gi coupling is observed, as in NTSR1, we performed 3D classifications focusing on the alignment of LPA₁ and G protein (lines 266 to 268).

Scientific concern:

1. The structure of the bound agonist (ONO-0740556) in the cryo-EM structure of the LPA1:agonist complex (Figure 1b) exhibits an unstable amide dihedral angle of -106.7 degrees (my appreciation to the authors for providing the structure to accompany this review so that the actual angle could be verified). Minimum energy conformations are expected to have amide dihedral angles near either 180 degrees or 0 degrees. Have the authors considered if a lower energy conformation of ONO-0740556 can be fitted into the cryo-EM map? If a more reasonable conformation cannot be fitted into the cryo_EM map, discussion of this high energy conformation in the context of the complex should be added to the manuscript.

According to the suggestion, we corrected the model of ONO-0740556 with the constraint that the amide dihedral angle is flat (left, original; right revised).

Minor errors to correct:

1. The structure of compound 5 in scheme 2 does not match the name of compound 5 defined in the supplemental methods. Compound 5 should be shown as the unprotected amine as the conditions defined for conversion of compound 5 to 6 should accomplish both the amine protection and the carboxylic acid reduction.

In our case, we used the protected amine as compound 5. Therefore, the conversion of compound 5 to 6 involves only the carboxylic acid reduction (ClCO₂i-Bu was only used to generate mixed carboxylic anhydrides *in situ*).

Reviewer #3

This manuscript by Akasaka et al reports a cryo-EM structure of human LPA1-Gi complex bound to a LPA analog ONO-0740556. LPA1 participates in nervous-system tissue development and chondrocyte differentiation, as a drug target in the treatment of cancer, inflammation, and neuropathic pain. The authors provided structural insights into ligand binding, receptor activation, and G protein coupling of LPA1. Structural determinants for lipids preference among LPA1, SIP3, and CB1, and rearrangements of motifs during LPA1 activation are identified. The resolution is relatively low (3.5Å), but the density around the ligand could be observed. Generally, this manuscript is not well written to support the authors claims with several missing information in the text and figures. This reviewer suggests major revisions before publication. The authors are strongly encouraged to edit the manuscript extensively to improve the text and figures.

We appreciate the reviewer's constructive evaluation of our paper.

1. The affiliations for all authors are missing in the manuscript.

According to the suggestion, we added the affiliations.

2. Line 188, the sentence "On the intracellular side, TM6 is displaced outward by about 11 Å, and TM7 is shifted inward by about 3.5 Å", it is better to point out which residues were used in the measurement.

According to the suggestion, we specified the residues used in the measurements in the legend. Moreover, since the distances are incorrect based on the previous model, we revised them with the current model (lines 190, 191, 393 and 394).

3. Line 435, what does the “100th volume” refer to?

We are sorry for our typo. “100th volume” should be “one 100th volume” (in new line 494).

4. Figure 2a, all residues interacting with the ligand shown in sticks should be labeled.

According to the suggestion, we have additionally labeled all of the residues.

5. Figure 2d, what are the red and blue boxes standing for?.

The colors indicated the charges of the residues. However, since they were confusing, we changed the colors to black.

6. Figure 2e and Figure S3, the authors are encouraged to provide the sequence alignment results, and color the residues to show if they are conserved or not.

According to the suggestion, the conserved residues in EDG family were colored red.

7. Figure 4d, all residues are missing..

According to the suggestion, some residues were added and all the residues were labeled.

8. Table S1, the Ca²⁺ influx assay should be represented as Mean ± S.E.M. from three independent experiments.

The NanoBiT-G-protein dissociation assay was performed instead of the Ca²⁺ influx assay, and the results are expressed as Mean ± S.E.M. from three independent experiments (Fig. 1b and Table S1).

9. The authors are encouraged to perform the Ca²⁺ influx assay for the mutagenesis on the ligand-binding site to validate the receptor-ligand interactions identified in the structure.

According to the suggestion, we performed the mutational analysis on the ligand-binding site. Since it is now difficult to perform the Ca²⁺ influx assay due to material and machinery problems, we instead performed more direct NanoBiT-G-protein dissociation assay for six residues that interact with the ligand (Figs. 2e and S3, Table S3).

10. *Figure 1c, what are the different contours?*

According to the suggestion, we changed the wording from the different contours to the different density levels (line 367).

11. *Figure S3 is a table and should be Table S3.*

According to the suggestion, we corrected them.

12. *Figure S4, the structure of LPA1 antagonist should be added.*

According to the suggestion, we added the structure of the LPA1 antagonist (Fig. S4a).

13. *Figure S6, which helices are shown in the lower panel?*

According to the suggestion, we labeled the helices.

14. *The title is “Structure and dynamics of Human Lysophosphatidic Acid Receptor 1”, but no dynamics results are presented. The authors are encouraged to perform the molecular dynamics simulations to better support the author’s claims.*

As the title was not consistent with the manuscript content, we changed the title to “Structure of the active Gi-coupled human Lysophosphatidic Acid Receptor 1 complexed with a potent agonist” according to the reviewers’ comments. MD simulation is beyond the scope of the manuscript.

REVIEWERS' COMMENTS

Reviewer #1 (Remarks to the Author):

The authors have done an outstanding job addressing the comments raised on the original version of the manuscript. I congratulate them on this wonderful study and recommend the publication of this manuscript without any further delay.

Arun K. Shukla, Ph.D.

Reviewer #2 (Remarks to the Author):

The authors present a valuable cryo-EM structure of the LPA1 receptor in complex with an agonist. Comparisons with active-state structures of other lipid receptors bound with agonists highlight differences in ligand recognition, and extensive analysis of polymorphisms in the receptor-Gi interface are also of interest. All recommendations from the previous submission have been addressed well by the authors.

Reviewer #3 (Remarks to the Author):

This reviewer is satisfied with the responses and revised manuscript.

Reviewer #1

The authors have done an outstanding job addressing the comments raised on the original version of the manuscript. I congratulate them on this wonderful study and recommend the publication of this manuscript without any further delay.

We appreciate the reviewer's favorable evaluation and recommendation of our paper.

Reviewer #2

The authors present a valuable cryo-EM structure of the LPA1 receptor in complex with an agonist. Comparisons with active-state structures of other lipid receptors bound with agonists highlight differences in ligand recognition, and extensive analysis of polymorphisms in the receptor-Gi interface are also of interest. All recommendations from the previous submission have been addressed well by the authors.”.

We appreciate the reviewer's favorable evaluation of our paper.

Reviewer #3

This reviewer is satisfied with the responses and revised manuscript.

We are happy that the reviewer's satisfied with our responses and revised manuscript.